# SASQ: Static Activation Scaling for Quantization-Aware Training in Large Language Models

## Abstract

Large language models (LLMs) excel at natural language tasks but face deployment challenges due to their growing size outpacing GPU memory advancements. Model quantization mitigates this issue by lowering weight and activation precision, but existing solutions face fundamental trade-offs: dynamic quantization incurs high computational overhead and poses deployment challenges on edge devices, while static quantization sacrifices accuracy. Existing approaches of quantization-aware training (QAT) further suffer from weight training costs. We propose **SASQ**: a lightweight QAT framework specifically tailored for activation quantization factors. SASQ exclusively optimizes only the quantization factors (without changing pre-trained weights), enabling static inference with high accuracy while maintaining deployment efficiency. SASQ adaptively truncates some outliers, thereby reducing the difficulty of quantization while preserving the distributional characteristics of the activations. SASQ not only surpasses existing SOTA quantization schemes but also outperforms the corresponding FP16 models. On LLaMA2-7B, it achieves 5.2% lower perplexity than QuaRot and 4.7% lower perplexity than the FP16 model on WikiText2.

## 1 Introduction

Large language models (LLMs) have demonstrated remarkable performance across a variety of tasks in natural language processing. The rapid parameter growth in LLMs (e.g., GPT-4's 1760B OpenAI et al. (2024)) far exceeds hardware memory capacity growth, drastically increasing deployment costs. Similarly unacceptable are the computational demands and communication latency from hardware stacking. Crucially, model sizes plateaued at around 1600B post-2022, demonstrating that GPU/TPU memory constraints now limit model advancement. One promising way to compress LLMs for lowing the costs of deployment is Quantization Dettmers et al. (2022); Yao et al. (2022). Quantization converts high-precision floats to low-precision integers, compressing redundant information to reduce storage and accelerate computation.

Current quantization techniques predominantly focus on Post-Training Quantization, which avoids model retraining and derives quantization parameters through predefined algorithms. However, these parameters may not inherently align with the functional relationships of weight or activation matrices, implying that computationally derived quantization parameters' values are not necessarily optimal. In contrast, Quantization-aware training (QAT) achieves a balance between quantization errors and weights by incorporating fake quantization operators during training to simulate quantization errors. However, constrained by the inherent limitations of Post-Training Quantization (PTQ) methods, large-scale models such as LLaMA-2-7B Touvron et al. (2023) necessitate dynamic quantization strategies to preserve performance, which typically employ dynamic computation of quantization factors and therefore incur significant computational overhead. Meanwhile, current Quantization-Aware Training approaches primarily focus on fine-tuning weights to accommodate quantization errors. LLM-QAT Liu et al. (2023) introduces a data-free distillation framework for quantization-aware training, enabling feasible optimization of quantized models without reliance on original training data. However, fine-tuning the weights introduces significant training overhead and may degrade the model's robustness. This paper proposes the Static Activation Scaling Quantization-Aware Training (SASQ) method, which incorporates pseudo-quantization operators

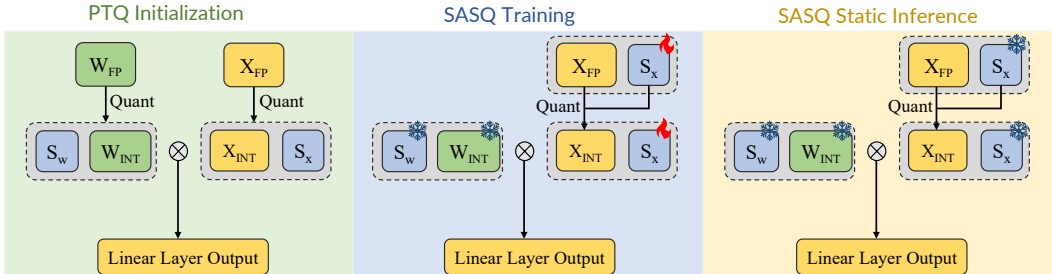

Figure 1: An overview of SASQ. First, PTQ with a calibration set is used to obtain the quantized model and initial quantization factors. During the SASQ Training stage, only the quantization factors are optimized. Finally, in the SASQ Static inference stage, these factors are directly applied for inference without online computation.

during training by leveraging the concept of quantization-aware training (QAT). Unlike conventional QAT methods, our approach specifically optimizes only activation quantization factors during training, rather than the large-scale weight parameters. Since no modification is introduced to the pre-trained weights, the inherent capabilities of the model remain unaffected. Once optimized, these factors enable efficient static quantization during inference. We validate our methodology through comprehensive QAT experiments across diverse large language models, with evaluation spanning multiple tasks and datasets. Our main contributions are summarized as follows:

1. We proposed a lightweight QAT and static quantization framework that substantially reduces training overhead and perplexity without changing weights. The SASQ quantization approach demonstrates superior performance on LLMs compared to state-of-the-art (SOTA) quantization schemes like QuaRot Ashkboos et al. (2024), AWQ Lin et al. (2024), and LLM-QAT Liu et al. (2023). Our experiments on the LLaMA-2-7B model with the WikiText2 Merity et al. (2016) dataset show that SASQ achieves better perplexity, which is 5.2%, 6.9%, 54.3%, and 4.7% lower than QuaRot, AWQ, LLM-QAT, and FP16 model, respectively.

2. Our method adaptively truncates certain outliers, preserving the overall distribution information of activations while reducing the difficulty of quantization, thereby enabling the quantized model to achieve improved performance. Experiment results suggest that specialized processing of input tensors during inference can further enhance model performance.

3. To address the particular challenge of error accumulation during autoregressive generation (e.g., in mathematical reasoning and code generation), we propose a novel phased quantization mechanism integrated with SASQ. This approach dynamically adjusts quantization granularity across different generation phases. When evaluated on the demanding mathematical reasoning benchmark (AIME2024), it preserves 87.5% of the FP16 model's accuracy. To our knowledge, we present the first quantization work that achieves such comprehensive preservation of mathematical capabilities in quantized large language models.

## 2 RELATED WORK

**Model Quantization.** Quantization techniques are principally divided into Quantization-Aware Training (QAT) and Post-Training Quantization (PTQ). The core distinction between them lies in whether retraining is needed during quantization.

LLM.int8() Dettmers et al. (2022) uses a mixed-precision approach, retaining FP16 precision for outlier activation values while quantizing the remaining values to INT8. This approach introduces additional floating-point computation overhead and increase the complexity of quantization and de-quantization operations. ZeroQuant Yao et al. (2022) employs dynamic per-token activation quantization and group-wise weight quantization, achieving its quantization through dynamic computational methods as well. SmoothQuant Xiao et al. (2023) employs pre-computed smoothing factors to transform both activations and weights, shifting the quantization difficulty from activation matrices (which typically contain significant outliers) to the more quantization-friendly weights. These meth-

ods use different ways to achieve quantization but cannot maintain accuracy in static quantization or need high cost dynamic activation quantization. Some recent works can achieve ultra-low-bits quantization while have good performance like QuaRot, PrefixQuant Chen et al. (2024). These works propose innovative approaches to cope with outliers in quantization, improving LLM performance after quantization Zhu et al. (2024). To further optimize the quantization error, some works achieve algorithm-hardware co-design like Olive Guo et al. (2023), Tender Lee et al. (2024). For the implementation of real integer-only quantized matrix multiplication, NVIDIA's CUTLASS INT8 GEMM kernels can be leveraged, which necessitate storing intermediate results in INT32 or FP32 precision.

**Quantization Aware Training.** As previously stated, Quantization-Aware Training (QAT) refers to a methodology that involves training the model during the quantization process to minimize the quantization error. Despite its substantial computational demands, limited QAT studies such as LLM-QAT, BitDistiller Du et al. (2024), and OneBit Xu et al. (2024) have provided valuable insights. While they incorporated training into the quantization process, it is noteworthy that this training was not specifically targeted at optimizing quantization factors, but rather focused on modifying the model architecture itself. Nevertheless, their work provided critical guidance for implementing gradient descent under circumstances where quantization might induce truncation effects.

## 3  PRELIMINARIES

**Large Language Models.**  Large Language Models (LLMs) are Transformer-based systems Vaswani et al. (2017). Each block computes attention (using Query, Key, Value matrices and softmax) and a Feed-Forward Network (FFN) with linear layers and activations. Quantization research primarily targets optimizing the linear transformations (matrix multiplications in attention and FFN layers) for efficient weight compression using fixed-point arithmetic, enabling full Transformer block computation. In this article, we use $\mathbf{X}*\mathbf{W}$ to represent matrix multiplication in linear layers of LLMs.

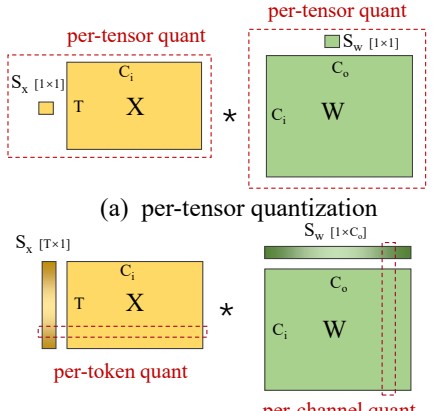

(a)  per-tensor quantization

(b) per-token + per-channel quantization

Figure 2: Schematic of quantization granularities (S represents quantization factor). Matrix regions are quantized using corresponding elements, marked by uniform bounding boxes. (a) per-tensor quantization: single factor for entire tensor, all elements in the matrix share the same quantization factor; (b) per-token + per-channel quantization: per-token means the elements in each row of the matrix share the same quantization factor; per-channel means the elements in each column of the matrix share the same quantization factor.

**Quantization.**  Quantization refers to the process of mapping high-precision numerical values (e.g., FP16 or FP32) to low-precision integer representations (e.g., INT8 or INT4). A common approach is linear quantization, which discretizes the dynamic range of values into a fixed number of integer levels. This method reduces redundancy in numerical representations, thereby accelerating computations and lowering memory footprint. A typical formula for symmetric quantization to an N-bit integer is as follows:

$$\mathbf{X}_{\text{int}} = \text{clamp}\left( \left\lfloor \frac{\mathbf{X}_{\text{fp}}}{\mathbf{S}} \right\rceil, -2^{N-1}, 2^{N-1} - 1 \right), \quad \mathbf{S} = \frac{\max(|\mathbf{X}|)}{2^{N-1} - 1} \tag{1}$$

where $X_{\text{int}}$ denotes the quantized integer representation of the original floating-point tensor $X_{\text{fp}}$. The clamp function ensures values are bounded within the $N$-bit signed integer range, while the scale factor $s$ is computed as the ratio of the maximum absolute value of $X$ to the maximum representable integer value. Asymmetric quantization introduces a zero-point offset to center the range around zero, saving one bit but increasing computational complexity. In this work, we adopt symmetric quantization for simplicity and balanced efficiency-performance trade-off.

The quantized computation can be expressed as follows:

$$\mathbf{Y} = \mathbf{X} \cdot \mathbf{W} = (\mathbf{S}_x \cdot \mathbf{X}_{\text{int}}) \cdot (\mathbf{W}_{\text{int}} \cdot \mathbf{S}_w) = \mathbf{S}_x \cdot (\mathbf{X}_{\text{int}} \cdot \mathbf{W}_{\text{int}}) \cdot \mathbf{S}_w \tag{2}$$

$$\mathbf{Y} = \mathbf{X} \cdot \mathbf{W} = \sum_{i=1}^{G} (\mathbf{S}_{x_i} \cdot \mathbf{S}_w) \cdot (\mathbf{X}_{\text{int},i} \cdot \mathbf{W}_{\text{int}}) \tag{3}$$

where $\mathbf{X}$, $\mathbf{W}$, $\mathbf{S}$, $G$ and $\mathbf{Y}$ denote the representations of the activation, weight, quantization factors, Group numbers and the final result, respectively. In block-wise matrix multiplication (especially when X is quantized in per-channel) the quantized computation can be formulated as a sequence of block-quantized matrix multiplications followed by accumulation. For accelerating computing GEMM (General Matrix Multiplication), we can use the corresponding INT matrix to multiply the kernel, first perform the INT matrix, and then perform inverse quantization to obtain the final floating-point result. In linear layers $\mathbf{Y} = \mathbf{X} \cdot \mathbf{W} + \mathbf{B}$, we omit quantizing bias $\mathbf{B}$ due to its negligible scale.

Quantization methods can be categorized by timing: static quantization computes scale factors offline using calibration data, while dynamic quantization calculates them during inference. By granularity, strategies include per-tensor, per-token, and per-channel quantization(see Figure 2). Per-tensor quantization maintains accuracy in some models but causes severe degradation in others with high activation outliers (e.g., LLaMA). Per-token/channel dynamic quantization preserves precision but adds computational overhead, while static methods still incur accuracy loss.

**Outlier.** Outliers are critical in quantized models. For value distributions with prominent outliers, quantization tends to disproportionately discard information from lower-magnitude values, degrading model performance. Since activation tensors generally exhibit larger outliers than weight matrices, suppressing activation outliers becomes essential for effective quantization. However, outliers present in activations carry information crucial to the model's reasoning process and significantly influence its output. Therefore, simply clipping these outliers can lead to performance degradation. Some studies attempt to mitigate this by shifting such outliers through mathematical transformations Xiao et al. (2023); Ashkboos et al. (2024), but these approaches inevitably alter the model weights, which can disrupt the delicate internal representations learned during pre-training Kumar et al. (2024). This limitation also applies to general QAT methods Liu et al. (2023); Du et al. (2024), which also require fine-tuning of weights.Motivated by this limitation, our work explores to reduce the impact of outliers on quantization through adaptive methods without modifying the original model weights.

# 4 SASQ FRAMEWORK

In this section, we propose a training-based method to derive static quantization factors, dubbed SASQ, is depicted in Figure1. We use per-channel quantization for both activation and weight, experimental results demonstrate that these trained quantization factors are generalizable and invariant to the input tokens during inference.

## 4.1 PTQ INITIALIZATION

Static quantization poses challenges due to the inability of pre-computed parameters to generalize effectively to unseen inputs. For instance, static quantization employing derived parameters had a perplexity exceeding 400 on the LLaMA2-7B model. We utilize the average of these pre-computed activation quantization parameters from the validation set as our initial quantization factors. Furthermore, prior to training, it is necessary to quantize the original model to introduce quantization errors during the training process. Finally, we initialize the quantized model with these quantization parameters and start training.

## 4.2 TRAINING

We incorporate differentiable fake quantization operators into the computational graph to enable end-to-end training. Our approach adopts straight-through estimation (STE) for gradient propagation through non-differentiable quantization operations : gradients for rounding operations are

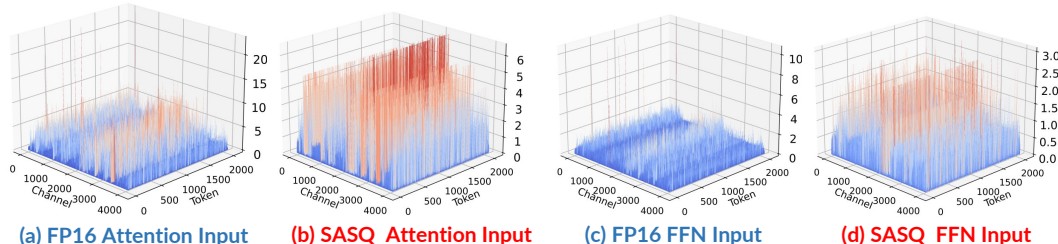

(a) FP16 Attention Input  (b) SASQ Attention Input  (c) FP16 FFN Input  (d) SASQ FFN Input

Figure 3: Comparison of Attention and FFN input activation distributions between FP16 and SASQ models in the middle (9th) linear module of LLaMA2-7B: SASQ reduces outliers through non-linear transformation.

directly set to 1, while gradients for clamped values outside the valid range are set to 0, and those within the range remain 1.

$$\frac{\partial \text{Round}(x)}{\partial x} = 1, \quad \frac{\partial \text{Clamp}(x)}{\partial x} = \begin{cases} 1 & \text{if } x \in [\text{min}, \text{max}] \\ 0 & \text{otherwise} \end{cases} \tag{4}$$

We found it more effective than complex nonlinear approximations like tanh or sigmoid, which introduce computational overhead and degrade performance. The training process is performed on the quantized model, implying that the gradient descent of the parameters inherently accounts for quantization errors during optimization.

SASQ directly incorporates only the numerical values of the quantization factors into the training process, utilizing perplexity minimization as the optimization objective. Unlike conventional QAT, which fine-tunes full model weights, our method exclusively optimizes these quantization factors without imposing explicit constraints on their update range, as symmetric quantization is adopted. This significantly reduces the training scale and memory footprint, allowing all factors to be trained jointly to guarantee a globally optimal solution. In contrast, existing block-wise fine-tuning methods are incompatible with our framework, as there is no guarantee that the combination of locally optimal values retains global optimality.

### 4.3 Effectiveness Analysis

The core operation of quantization can be expressed by the following formula:

$$\mathbf{Y} = \mathbf{X} \cdot \mathbf{W} = \mathbf{S}_x \cdot \mathbf{X}_{\text{int}} \cdot \mathbf{W} = \mathbf{X}' \cdot \mathbf{W} \tag{5}$$

$$\mathbf{Y} = \mathbf{X}' \cdot \mathbf{W} = \mathbf{X} \cdot \mathbf{I}' \cdot \mathbf{W} = \mathbf{X} \cdot f(\mathbf{W}) = g(\mathbf{X}) \cdot \mathbf{W} \tag{6}$$

This means that for the activation $\mathbf{X}$, we can perform a transformation, such that it essentially changes from $\mathbf{X}$ to a certain floating-point matrix $\mathbf{X}'$. The weights (and bias) require no analysis since they remain fixed whether quantized or not. To distinguish SASQ from conventional weight fine-tuning, we clarify the following: even for the same scaling factor in static quantization, the matrix $\mathbf{I}'$ generated by different activations $\mathbf{X}$ varies (i.e. $f(\mathbf{W})$ can't be linear function). This assert is attributable to the nonlinear characteristics of the quantization workflow (quant, round, truncation and dequant), implying that the matrix $\mathbf{I}'$ cannot be directly multiplied by the weight matrix $\mathbf{W}$ to form a fine-tuned weight matrix. In Table 3 SASQ (FP weights) modifies only activation tensors through quantization, rounding, truncation, and dequantization while keeping weights floating-point. Results surpass baseline as well, proving SASQ's efficacy stems solely from activation processing.

Compared to complex mathematical transformations for mitigating outliers (such as the transformation matrices introduced in SmoothQuant and QuaRot), we hypothesize that certain learned non-linear transformations might achieve a similar effect. However, our experiments show that such transformations can be realized simply by adjusting the quantization factors. The parameters of these nonlinear functions are learned through training, which implies that the functions preserve the optimal strategy for handling outliers: they effectively mitigate the impact of outliers to quantization while retaining essential information from the value distribution. As demonstrated in Table 3, the clamp operation plays a critical role in SASQ: the combination of the learned quantization factors

and the clamp function effectively truncates some outliers, thereby improving the performance of the quantized model. This can also be observed in Figure 3.

We can draw the following conclusion from both experiments and theory inference: **Quantization nonlinearly alters linear layers' inputs by shifting activation distributions (truncate outliers)—fundamentally distinct from weight fine-tuning.**

## 4.4 PHASED QUANTIZATION

Our experiments have validated the effectiveness of SASQ on language modeling and multiple-choice reasoning tasks. However, these tasks primarily rely on computation in the prefill stage. In practical applications such as code generation, dialogue, and mathematical problem solving, models must operate in the autoregressive generation stage, which poses new challenges for preserving accuracy after quantization. As an exploratory study, this section aims to provide a preliminary assessment of the potential of SASQ in more challenging open-ended generation tasks. To address the characteristics of the generation stage, we propose a phased quantization mechanism (Figure 4) and evaluate it using mathematical reasoning as a case study.

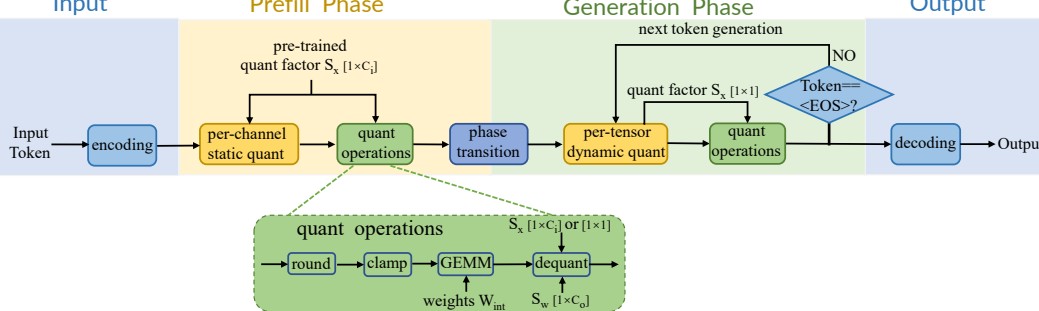

Figure 4: Phased Quantization Schematic for generation tasks. Prefill phase: per-channel static quantization; Generation phase: per-tensor dynamic quantization. GEMM represents the general matrix multiplication between activation and weights. EOS represents the token that signifies the termination of a generated sequence.

Prefill Phase Quantization Strategy: In the prefill stage, the large number of input tokens can be well matched with the per-channel factors learned during training, since SASQ derives the optimal quantization factor for each channel from multi-token matrices. Therefore, SASQ per-channel static quantization can be preserved in this stage.

Generation Phase Quantization Strategy: The generation phase leverages KV Cache optimization mechanisms, where each iteration processes only the newly generated token (i.e., single-row matrix operations). Conventional quantization factors optimized for large-scale input matrices (thousands of tokens) become suboptimal under these drastically reduced dimensionality conditions. Moreover, continuing per-channel quantization during the generation phase would significantly increase floating-point operations, as the number of generated tokens may exceed the input tokens by several dozen times. We implement dynamic per-token quantization that demonstrates operational equivalence to per-tensor approaches in this context, with acceptable computational overhead due to the single-token processing paradigm.

## 4.5 SUMMARY

A critical challenge in large language models quantization lies in determining optimal quantization factors to preserve model performance post-quantization. Our proposed SASQ method directly optimizes quantization factors through global training (as opposed to block-wise optimization), ensuring convergence to theoretically optimal values for static quantization. Therefore, SASQ provides an efficient pathway to derive these globally optimal static quantization factors. Notably, SASQ provides an additional advantage: unlike dynamic quantization during inference, it enables direct application

of pre-trained static quantization factors, thereby reducing computational latency. This feature is particularly beneficial for edge devices without floating-point units (GPU, NPU).

# 5 EXPERIMENTS

## 5.1 SETUPS

**Baselines.** We conduct comprehensive comparisons with FP16 baselines and representative quantization methods, including static/dynamic SmoothQuant and state-of-the-art approaches (AWQ, LLM-QAT, QuaRot, SpinQuant Liu et al. (2024)). The specific compared methods vary by models, with complete results provided in Table 1. We evaluate our W8A8 quantization method (8-bits weights and activations) on a range of models from HuggingFace, implemented in PyTorch. To evaluate the potential of the SASQ method in open-ended generation tasks, we apply the phased SASQ approach to mathematical reasoning tasks for a case study.

**Models and Datasets.** We implement SASQ on the LLaMA1/2/3 model and Qwen2.5 families. For the LLaMA2-13B and LLaMA3-8B models, the SASQ framework was applied to the floating-point models following the smoothing process Xiao et al. (2023). All training in our experiments is conducted solely on the WikiText103 Magnusson et al. (2023) training split. Evaluation is performed on the WikiText2 test split, the WikiText103 test set, and the PTB3 Marcus et al. (1994) test set to assess generalization capabilities. For SASQ method, we further analyze its performance on six zero-shot classification tasks: HellaSwag Zellers et al. (2019), ARC-Challenge, ARC-Easy Clark et al. (2018), COPA Roemmele et al. (2011), PIQA Bisk et al. (2020), and WinoGrande Sakaguchi et al. (2019). For generation tasks, we focus specifically on mathematical reasoning, evaluating our approach on the MATH-500 Lightman et al. (2023) and AIME2024 of America (2024) benchmarks to assess its capability in complex mathematical problem-solving.

**Implementation** Initial per-channel quantization scales are derived from a subset of the Pile validation set Gao et al. (2020). We employ Adam Kingma (2014) optimizers with a learning rate of $2 \times 10^{-4}$ for 6 epochs. Quantization-aware training initializes parameters using static quantization factors, with perplexity computed over validation samples serving as the loss function. During training, we incorporate only the quantization factors of the linear layers' activations as optimizable parameters. Since we adopt symmetric quantization, no constraints are applied to the update range of these parameters. The sequence length is set to 2048. The quantization granularity of weights is always set to per-channel. For general tasks, we apply per-channel quantization to activations. For mathematical reasoning, we implement per-channel quantization for activation in prefill stage, and per-tensor dynamic quantization for activation in generation stage. In generation configurations, the maximum number of tokens is set to 32,768[1], using sampling with temperature 0.6 and top-p value 0.95. Mathematical reasoning evaluation is performed via OpenCompass OpenCompass Contributors (2023), while multi-classification tasks are evaluated using the lm-eval-harness Gao et al. (2024) framework.

## 5.2 QUANTIZATION RESULTS

### 5.2.1 RESULTS ON DIFFERENT QUANTIZATION METHODS

In Table 1, we observe that the quantized inference results of SASQ actually outperform the floating-point model across all six models tested in our experiments. However, the results of PTQ quantization like SmoothQuant or QuaRot reveal that merely performing precise quantization on the model is insufficient to achieve better performance. In Table 1, SASQ's performance on the WikiText2 or PTB3 datasets rules out the influence of WikiText103 data participation in training. on LLaMA2-7B, SASQ reduced perplexity on WikiText2 by 4.7% (surpassing existing SOTA results) while maintaining 97.4% of the floating-point model's accuracy on multi-class classification tasks on average. Although SASQ was not specifically trained for multi-class tasks, all six models still maintained comparable performance to their floating-point counterparts. This demonstrates the superior robustness of the SASQ method.

---

[1]Qwen2.5-MATH has a maximum token limit of 4096 for its generated outputs.

Table 1: Comprehensive evaluation of quantization methods across multiple models and datasets. Perplexity(lower is better) is reported for WikiText2, PTB3, WikiText103 and their average (Avg). Accuracy(higher is better) is reported for HellaSwag, PIQA, COPA, ARC_C, ARC_E, WinoGrande tasks and their average (Avg). "-" indicates that the result is either not reported in public papers, or the baseline performance on certain benchmarks differs significantly (by more than 5%) from ours.

| Model | Method | Perplexity(↓) | | | | Accuracy(↑) | | | | | | |
|---|---|---|---|---|---|---|---|---|---|---|---|---|
| | | Wiki2 | PTB3 | Wiki103 | Avg. | HS | PQ | CP | A_C | A_E | WG | Avg. |
| LLaMA1-7B | **Baseline (FP16)** | **5.683** | **8.576** | **5.762** | **6.674** | **76.20** | **79.16** | **84.0** | **44.71** | **72.93** | **69.93** | **71.39** |
| | SMQ Dynamic | 5.716 | 8.620 | 5.796 | 6.711 | 75.81 | 78.84 | 87.0 | 44.54 | 75.21 | 70.24 | 71.77 |
| | AWQ | 5.78 | - | - | - | - | 77.97 | - | 41.21 | - | 68.75 | - |
| | LLM-QAT | 10.3 | - | - | - | 75.6 | 79.4 | - | - | 72.3 | 69.7 | - |
| | SASQ | *5.182* | *9.073* | *5.191* | *6.482* | 74.81 | 78.13 | 86.00 | 44.03 | 72.22 | 68.75 | 70.32 |
| LLaMA2-7B | **Baseline (FP16)** | **5.473** | **27.780** | **6.783** | **13.345** | **75.99** | **79.11** | **87.0** | **46.25** | **74.58** | **69.14** | **72.18** |
| | SMQ Dynamic | 5.510 | 27.994 | 6.828 | 13.444 | 76.10 | 77.80 | 88.0 | 46.15 | 74.10 | 68.58 | 71.79 |
| | SMQ Static | 448.524 | 863.796 | 387.995 | 566.772 | 33.55 | 60.07 | 85.0 | 27.56 | 52.15 | 64.48 | 53.80 |
| | QuaRot | 5.50 | - | - | - | 75.80 | 78.94 | - | 45.39 | 74.79 | 68.67 | - |
| | AWQ | 5.60 | - | - | - | - | 77.31 | - | 43.86 | - | 68.98 | - |
| | LLM-QAT | 11.4 | - | - | - | 74.00 | 78.20 | - | - | 73.60 | 67.70 | - |
| | SpinQuant | 5.70 | - | - | - | 74.80 | 78.90 | - | - | 74.00 | 68.90 | - |
| | SASQ | *5.214* | *12.581* | *5.492* | *7.870* | 73.46 | 77.86 | 88.0 | 44.45 | 72.22 | 67.72 | 70.32 |
| LLaMA2-13B | **Baseline (FP16)** | **4.884** | **36.621** | **7.099** | **16.201** | **79.37** | **80.52** | **91.00** | **49.06** | **77.40** | **72.30** | **74.94** |
| | SMQ Dynamic | 4.927 | 37.738 | 7.204 | 16.623 | 79.25 | 80.58 | 91.00 | 48.81 | 77.02 | 71.27 | 74.65 |
| | QuaRot | 4.90 | - | - | - | 79.38 | 80.36 | - | 49.15 | 77.31 | 71.98 | - |
| | AWQ | 4.97 | - | - | - | - | 79.00 | - | 47.70 | 79.04 | 73.24 | - |
| | LLM-QAT | 14.5 | - | - | - | 77.4 | 80.0 | - | - | 75.3 | 71.6 | - |
| | SASQ | *4.681* | *16.361* | *5.150* | *8.731* | 76.30 | 78.51 | 86.00 | 49.32 | 74.58 | 70.64 | 72.56 |
| LLaMA3-8B | **Baseline (FP16)** | **6.140** | **9.992** | **6.259** | **7.464** | **79.16** | **80.79** | **89.00** | **53.41** | **77.78** | **72.85** | **75.50** |
| | SMQ Dynamic | 6.243 | 10.128 | 6.365 | 7.579 | 78.95 | 80.09 | 89.00 | 52.39 | 77.10 | 72.45 | 75.00 |
| | LLM-QAT | 7.7 | - | - | - | 76.0 | 79.0 | - | - | 77.6 | 72.4 | - |
| | SpinQuant | 6.50 | - | - | - | 78.10 | 79.60 | - | - | 76.50 | 72.40 | - |
| | SASQ | *6.117* | *10.594* | *6.126* | *7.612* | 77.73 | 79.27 | 87.00 | 51.88 | 77.86 | 72.06 | 74.30 |
| Qwen2.5-1.5B | **Baseline (FP16)** | **9.270** | **15.429** | **9.149** | **11.283** | **67.62** | **75.68** | **82.00** | **44.62** | **71.51** | **62.59** | **67.00** |
| | SMQ Dynamic | 9.425 | 15.693 | 9.283 | 11.467 | 67.44 | 75.46 | 83.00 | 43.09 | 69.57 | 64.33 | 67.15 |
| | SASQ | *8.465* | *16.920* | *8.227* | *11.204* | 65.36 | 74.65 | 77.00 | 44.03 | 71.55 | 65.04 | 66.27 |
| Qwen2.5-7B | **Baseline (FP16)** | **6.850** | **11.286** | **6.878** | **8.338** | **78.71** | **80.03** | **91.00** | **51.11** | **77.78** | **71.82** | **75.74** |
| | SMQ Dynamic | 6.994 | 11.503 | 7.013 | 8.503 | 78.49 | 79.65 | 88.00 | 48.89 | 75.38 | 71.27 | 73.61 |
| | SASQ | *6.526* | *12.957* | *6.439* | *8.641* | 75.71 | 78.18 | 87.00 | 49.74 | 76.47 | 71.19 | 73.38 |

### 5.2.2 RESULTS ON QUANTIZATION FOR MATHEMATICS

In Table 2, SASQ demonstrated good performance preservation across both autoregressive modeling and mathematical reasoning tasks. For DeepSeek-R1-Distill-Qwen2.5-MATH-1.5B model, we observed a 2.1× performance enhancement on the WikiText2 benchmark, while maintaining nearly 80% of the MATH-500 and 87.5% of the AIME2024 mathematical reasoning capability. It is particularly noteworthy that when employing either truncated token-length static per-token quantization during the prefill phase or maintaining identical per-channel static quantization throughout the generation phase, the results were observed to approach zero. Additional experimental results on these models see Appendix.

Table 2: Perplexity and MATH accuracy results of DeepSeek-Distilled-1.5B and Qwen2.5-MATH-1.5B on various datasets. The accuracy of the Naive Static represents the best performance among static methods, including SMQ-static and SASQ-static.

| Model | Dataset | Metric | FP16 | Naive Static | SASQ (Phased) | Rate (Quant/Float) |
|---|---|---|---|---|---|---|
| DeepSeek-Distilled-1.5B | WikiText2 | ppl(↓) | 40.828 | – | 19.518 | 1 / **2.09** |
| | MATH-500 | acc(↑) | 81.2% | 3.0% | 64.8% | 79.8% |
| | AIME2024 | acc(↑) | 26.6% | 0 | 23.3% | 87.5% |
| Qwen2.5-MATH-1.5B | WikiText2 | ppl(↓) | 17.685 | – | 15.415 | 1 / **1.15** |
| | MATH-500 | acc(↑) | 58.6% | 1.4% | 44.0% | 75.1% |
| | AIME2024 | acc(↑) | 6.67% | 0 | 6.67% | 100.0% |

## 5.3 EFFICIENCY AND PERFORMANCE ANALYSIS

SASQ demonstrates substantial advantages in training efficiency compared to conventional QAT approaches. For instance, LLM-QAT requires over 100k training samples, hundreds of hours for data generation, and additional hundreds of hours for training under identical single-GPU configurations, making deployment prohibitively costly. In contrast, SASQ trains LLaMA2-13B model in approximately 15 hours using only two A800-80GB GPUs and relying solely on the WikiText103 dataset, thereby drastically reducing both data requirements and training overhead. In terms of inference latency, SASQ also outperforms dynamic quantization methods. Additional speedup and Memory Saving results on these models see Appendix. Figure 5 presents the training curves under different learning rates and initial quantization factors. The experimental results indicate that SASQ achieves stable convergence and exhibits low sensitivity to initialization.

## 5.4 ABLATION STUDY

To investigate why quantized models achieve lower perplexity than their floating-point counterparts, we conducted ablation studies on core quantization operations and quantized weights. Our findings reveal that the clamp function—not the rounding function—is the decisive factor driving this result. This result has been validated across multiple models and datasets, we take LLaMA1/2-7B for example(Table 3):

The experimental results and our analysis suggest the existence of an underlying mechanism through which non-linear transformations applied to input activation tensors may enhance model performance. Analogous to our findings, PrefixQuant Chen et al. (2024) reduces activation outliers by prepending specialized tokens to input sequences. Crucially, we posit that applying low-cost transformations to activation tensors can potentially enhance model performance with negligible computational overhead.

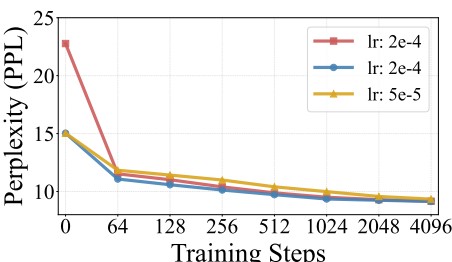

Figure 5: Training convergence under different initialization values of quantization factors and varying learning rates. The loss function is defined as the perplexity over the first five samples, which is equivalent to the PPL shown in the figure.

Table 3: Perplexity results across multiple datasets for LLaMA 1/2-7B models under different configurations of SASQ. 'w/o' denotes 'without', and 'FP weights' indicates that the weights are kept in full precision while the activations are quantized and dequantized with the pre-trained factors.

| | | LLaMA2-7B | | | | | LLaMA1-7B | | |
|---|---|---|---|---|---|---|---|---|---|
| Method | Variant | Perplexity (↓) | | | Method | Variant | Perplexity (↓) | | |
| | | Wiki2 | PTB3 | Wiki103 | | | Wiki2 | PTB3 | Wiki103 |
| FP16 | Baseline | 5.47 | 27.78 | 6.78 | FP16 | Baseline | 5.68 | 8.58 | 5.76 |
| SASQ | Full | 5.214 | 12.581 | 5.492 | SASQ | Full | 5.182 | 9.073 | 5.191 |
| | w/o round | 5.241 | 13.051 | 5.581 | | w/o round | 5.232 | 9.011 | 5.252 |
| | w/o clamp | **5.467** | **26.06** | **6.747** | | w/o clamp | **5.679** | **8.638** | **5.753** |
| | FP weights | 5.214 | 12.633 | 5.493 | | FP weights | 5.184 | 9.081 | 5.194 |

## 6 CONCLUSION

We propose SASQ method, which derives static quantization parameters by applying quantization-aware training (QAT) to calibration factors, demonstrating superior performance across various benchmarks. SASQ eliminates the real-time computation overhead of dynamic quantization while enabling efficient deployment on chips lacking floating-point units. SASQ enhances the performance of quantized models by applying nonlinear activation transformations (adaptively truncates some outliers), enabling them to achieve good performance on many tasks.

# 7 ETHICS STATEMENT

All the authors have know and acknowledge all the General Ethical Principles. Including but not limited to: Contribute to society and to human well-being. Uphold high standards of scientific excellence. Avoid harm. Be honest, trustworthy and transparent. Be fair and take action to avoid discrimination. Respect the work required to produce new ideas and artefacts. Respect privacy. Honour confidentiality. We declare that this work complies with all ethical standards.

# 8 REPRODUCIBILITY STATEMENT

The experimental results presented in this paper demonstrate strong reproducibility. However, as the paper has not yet been formally published, the complete reproduction code cannot be fully disclosed at this time. In the supplementary materials, we provide only a portion of the quantization-aware training code necessary (train.py) for reproducibility checks and review assistance. We commit to releasing all code and optimal scale file .pth upon paper acceptance.

All datasets used are publicly available within the research domain. The data processing scripts for perplexity evaluation can be found within the training code, as the training loss function is essentially the perplexity calculated on the first five samples. Additionally, processing scripts for other tasks are available on the platforms referenced in the main text.

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

# A APPENDIX

## A.1 STATEMENT ON THE USE OF LLMS

The LLMs were used to assist the authors in language polishing and structural adjustment of the manuscript.

## A.2 DEQUANT STRATEGY OF PER CHANNEL QUANTIZATION

There are many methods to achieve dequantization, we provide one for reference here:

$$\mathbf{Y} = \mathbf{X} \cdot \mathbf{W} = \sum_{i=1}^{G} (\mathbf{S}_{x_i} \cdot \mathbf{S}_w) \cdot (\mathbf{X}_{\text{int},i} \cdot \mathbf{W}_{\text{int}}) = \left( \sum_{i=1}^{G} \mathbf{S}_{x_i} \cdot (\mathbf{X}_{\text{int},i} \cdot \mathbf{W}_{\text{int}}) \right) \cdot \mathbf{S}_w \qquad (7)$$

This strategy can effectively implement the dequantization stage of per-channel quantization described in the paper. Taking $\mathbf{X}$ with shape $[\mathbf{T}, \mathbf{C}_i]$ and $\mathbf{W}$ with shape $[\mathbf{C}_i, \mathbf{C}_o]$ as an example, $\mathbf{S}_x$ has shape $[1, \mathbf{C}_i]$ and $\mathbf{S}_w$ has shape $[1, \mathbf{C}_o]$. The partitioned column vector $\mathbf{X}_{\text{int}}$ $[\mathbf{T}, 1]$ and row vector $\mathbf{W}_{\text{int}}$ $[1, \mathbf{C}_o]$ are multiplied to obtain an INT matrix $[\mathbf{T}, \mathbf{C}_o]$, with a total of Ci such matrices being accumulated. During accumulation, each INT matrix is multiplied by the corresponding quantization factor in $\mathbf{S}_x$ and then accumulated to obtain a floating-point matrix $[\mathbf{T}, \mathbf{C}_o]$. This matrix can be directly multiplied by $\mathbf{S}_w$ since their second dimensions match, resulting in the dequantized matrix. The total number of multiplications or additions in this dequantization strategy is consistent with that of conventional dequantization strategies.

## A.3 MORE EXPERIMENTS CONFIGS

Initial scaling factors are derived from a subset of The Pile validation set. The quantization-aware training process initializes parameters using static quantization factors obtained from the validation set, while perplexity computed across 5 validation samples serves as the loss function for optimization. Model parallelism is employed to enable multi-GPU training. The learning rate is scheduled with a linear warmup strategy, and gradient clipping is applied when gradients exceeded a specified threshold.

Table 4: Evaluation of quantization methods on DeepSeek Distilled and Qwen2.5-MATH models. Perplexity(lower is better) is reported for WikiText2, PTB3, WikiText103 and their average (Avg). Accuracy(higher is better) is reported for HellaSwag, PIQA, COPA, ARC_C, ARC_E, WinoGrande tasks and their average (Avg).

| Model | Method | Perplexity(↓) | | | | Accuracy(↑) | | | | | | |
|---|---|---|---|---|---|---|---|---|---|---|---|---|
| | | Wiki2 | PTB3 | Wiki103 | Avg. | HS | PQ | CP | A_C | A_E | WG | Avg. |
| DeepSeek-Distilled-Qwen2.5-MATH-1.5B | **Baseline (FP16)** | **40.828** | **60.261** | **39.063** | **45.722** | **44.75** | **65.78** | **64.0** | **34.64** | **56.06** | **55.56** | **53.13** |
| | SMQ Dynamic | 42.113 | 62.134 | 40.537 | 47.007 | 44.19 | 64.96 | 65.0 | 33.36 | 55.17 | 56.59 | 53.21 |
| | SMQ Static | 62.672 | 84.021 | 58.050 | 67.205 | 40.72 | 62.13 | 59.0 | 31.06 | 52.02 | 52.41 | 49.56 |
| | SASQ | *19.518* | 38.283 | 18.050 | 28.209 | 42.43 | 63.39 | 63.0 | 30.12 | 57.45 | 54.14 | 51.76 |
| | SASQ(FP weights) | 19.502 | 38.318 | 18.039 | 28.191 | 42.25 | 63.17 | 63.0 | 30.12 | 51.26 | 53.99 | 50.63 |
| DeepSeek-Distilled-Qwen2.5-MATH-7B | **Baseline (FP16)** | **25.129** | **33.925** | **24.318** | **28.861** | **60.76** | **72.42** | **75.0** | **44.20** | **66.88** | **60.54** | **63.30** |
| | SMQ Dynamic | 25.296 | 33.733 | 24.438 | 28.895 | 59.63 | 71.16 | 70.0 | 43.85 | 65.86 | 58.40 | 61.48 |
| | SMQ Static | 32.485 | 43.784 | 31.737 | 37.506 | 53.73 | 68.34 | 68.0 | 39.33 | 58.0 | 57.62 | 57.50 |
| | SASQ | *13.313* | 26.371 | 12.520 | 19.627 | 54.59 | 67.47 | 67.0 | 40.10 | 59.85 | 60.06 | 58.18 |
| | SASQ(FP weights) | 13.301 | 26.348 | 12.506 | 19.609 | 54.83 | 67.08 | 67.0 | 39.25 | 59.34 | 58.48 | 57.66 |
| Qwen2.5-MATH-1.5B | **Baseline (FP16)** | **17.685** | **27.547** | **16.901** | **20.711** | **49.79** | **68.66** | **62.0** | **40.70** | **64.14** | **57.93** | **57.20** |
| | SMQ Dynamic | 17.891 | 27.902 | 17.083 | 20.959 | 49.40 | 68.11 | 69.0 | 39.84 | 64.18 | 56.85 | 57.92 |
| | SMQ Static | 19.800 | 30.876 | 18.878 | 23.185 | 48.11 | 65.78 | 60.0 | 37.80 | 61.99 | 55.09 | 54.80 |
| | SASQ | *15.415* | 30.347 | 14.406 | 20.056 | 45.41 | 64.86 | 63.0 | 34.65 | 58.34 | 55.88 | 53.69 |
| Qwen2.5-MATH-7B | **Baseline (FP16)** | **11.580** | **16.806** | **11.231** | **13.206** | **65.45** | **74.37** | **77.0** | **49.32** | **72.56** | **63.93** | **67.11** |
| | SMQ Dynamic | 11.766 | 17.114 | 11.408 | 13.429 | 64.66 | 74.53 | 77.0 | 49.32 | 73.19 | 64.56 | 67.21 |
| | SMQ Static | 14.366 | 23.098 | 13.930 | 17.131 | 62.05 | 71.33 | 78.0 | 46.67 | 70.03 | 60.69 | 64.80 |
| | SASQ | *10.454* | 20.000 | 9.931 | 13.462 | 60.99 | 69.26 | 77.0 | 47.27 | 70.03 | 63.93 | 64.75 |

## A.4 ADDITIONAL EXPERIMENTS RESULTS

### A.4.1 ADDITIONAL QUANTIZATION RESULTS

Supplementary Table 5 presents additional experimental results. As shown, the SASQ method continues to outperform original full-precision models across these architectures. However, as these models are inherently mathematical-domain models, the finding that quantized models achieve lower perplexity than their floating-point counterparts is therefore more pronounced. We further validate these findings on both DeepSeek-Distilled-Qwen2.5-MATH and Qwen2.5-MATH models, confirming our inference: The SASQ approach enhances model performance by non-linearly altering the distribution of input activations rather than through weight fine-tuning.

### A.4.2 ADDITIONAL SPEEDUP AND MEMORY SAVING RESULTS

On wiki103 with 200 samples, SASQ achieves 3.5% (1.5B) and 1.7% (7B) speedups over SMQ-dynamic under INT8 quantization (Table 5) on Qwen2.5 model. This improvement arises from SASQ's use of precomputed quantization factors, whereas SMQ-dynamic requires runtime recalculation. However, the performance gap tends to diminish as inference time increases. Finally, SASQ achieves competitive weight compression ratios (with biases kept in floating-point). Specifically, Qwen2.5-1.5B and 7B reach 63.33% and 57.23% compression, respectively, while LLaMA1-7B/LLaMA2-7B obtain 52.03% owing to their bias-free linear layer design. The compressed LLaMA3-8B model retains 56.6% of the FP16 model size, while LLaMA2-13B retains 51.3%.

Table 5: Inference Latency on Wiki103 Set and Memory Saving results. All inference was performed on NVIDIA-V100 GPU and Qwen2.5 model.

| Size | Latency (s) (↓) | | | | Memory (GB) (↓) | | |
|------|------|-------|------|---------|------|------|--------|
| | FP16 | SMQ-D | SASQ | Speedup | FP16 | SASQ | Saving |
| 1.5B | 170 | 118 | 114 | 1.48× | 3.31 | 2.09 | 1.58× |
| 7B | 667 | 481 | 474 | 1.41× | 14.19 | 8.12 | 1.75× |

## A.5 ADDITIONAL ABLATION STUDY RESULTS

As shown in Table 6, the results on Qwen2.5 models also indicate that the fundamental factor responsible for the performance surpassing the floating-point model in SASQ is the clamp function, which enhances model performance by truncating partial outliers.

## A.6 RESULTS SOURCE OF OTHER METHOD

For LLaMA 1/2/3 models, perplexity and zero-shot task results of the LLM-QAT method are sourced from the LLM-QAT paper Liu et al. (2023) and SpinQuant Liu et al. (2024). Perplexity and zero-shot task results of the SpinQuant method are sourced from SpinQuant paper. Perplexity results of the AWQ method are derived from the AWQ paper Lin et al. (2024), while its zero-shot task results originate from the AutoRound GitHub repository. For LLaMA-2 models, both perplexity and zero-shot results of the QuaRot method are obtained from the QuaRot paper Ashkboos et al. (2024).

Table 6: Perplexity results across multiple datasets for Qwen2.5 models under different configurations of SASQ.

| Qwen2.5-7B | | | | | Qwen2.5-1.5B | | | | |
|---|---|---|---|---|---|---|---|---|---|
| Method | Variant | Perplexity (↓) | | | Method | Variant | Perplexity (↓) | | |
| | | Wiki2 | PTB3 | Wiki103 | | | Wiki2 | PTB3 | Wiki103 |
| FP16 | Baseline | 6.850 | 11.286 | 6.878 | FP16 | Baseline | 9.270 | 15.429 | 9.149 |
| SASQ | Full | 6.526 | 12.957 | 6.439 | SASQ | Full | 8.465 | 16.920 | 8.227 |
| | w/o round | 6.505 | 12.790 | 6.429 | | w/o round | 8.402 | 16.759 | 8.170 |
| | w/o clamp | **6.870** | **11.376** | **6.891** | | w/o clamp | **9.339** | **15.588** | **9.213** |

