# OpenReview forum: "SASQ: Static Activation Scaling for Quantization-Aware Training in Large Language Models"
_ICLR.cc/2026/Conference — ICLR 2026 Conference Withdrawn Submission_

### Official Review · Reviewer_GYVV · 2025-10-25

**Soundness:** 2
**Presentation:** 2
**Contribution:** 2
**Rating:** 4
**Confidence:** 4

**Summary:**

The paper proposes SASQ (Static Activation Scaling Quantization-aware Training) — a lightweight quantization-aware training (QAT) framework for large language models (LLMs).
Unlike prior QAT methods that fine-tune model weights, SASQ only optimizes activation quantization factors, keeping the pre-trained weights fixed. This enables static quantization (low inference cost) with accuracy comparable to or even better than FP16 baselines.
It introduces a phased quantization for autoregressive generation, using static per-channel quantization in the prefill phase and dynamic per-token quantization during generation.

**Strengths:**

Focusing QAT solely on activation scaling factors is conceptually elegant and avoids expensive weight fine-tuning.

**Weaknesses:**

The paper explains why SASQ works mainly empirically. A more formal analysis of why optimizing scaling factors alone can work would strengthen the contribution.

The paper lacks validation on instruction-following models, which are essential for evaluating practical performance and generalization.

The baselines compared in this paper are mainly LLM-QAT and SpinQuant, both of which are relatively early methods.

**Questions:**

What is the calibration dataset used in this paper?

The paper raises some factual concerns. For example:

“The rapid parameter growth in LLMs (e.g., GPT-4’s 1760B; OpenAI et al., 2024) far exceeds hardware memory capacity growth, drastically increasing deployment costs.”

How is the number “1760B parameters” for GPT-4 obtained? Is this figure accurate or supported by any official source?

Similarly, the statement

“Crucially, model sizes plateaued at around 1600B post-2022, demonstrating that GPU/TPU memory constraints now limit model advancement.”

also lacks clear evidence. How was “1600B” determined? Are there reliable data or citations supporting this claim?

---

> ### Author Response · Authors · 2025-11-27
>
> We extend our sincere gratitude to the reviewers for their thorough review and encouraging feedback on our work. The following points aim to address the concerns and questions raised:
>
> Weakness1: “The paper explains why SASQ works mainly empirically. A more formal analysis of why optimizing scaling factors alone can work would strengthen the contribution.”
> The phenomenon where the quantized model outperforms its full-precision counterpart is indeed counter-intuitive. Our primary explanation in the paper is that the learned quantization factors, in conjunction with the clamp function, adaptively truncate outlier activations, leading to the superior performance of the quantized model. Our ablation studies confirm the crucial role of the clamp function. Furthermore, observations from Figure 3 in the inference process visually demonstrate the significant effect of this adaptive truncation on handling outliers.
>
> Weakness2: “The paper lacks validation on instruction-following models, which are essential for evaluating practical performance and generalization.”
> We understand the concern regarding validation on instruction-following models. However, this has not been a primary focus in the mainstream literature of model quantization, as evidenced by established works like QuaRot and SmoothQuant. That said, recognizing the paramount importance of generative models in the LLM domain, we have introduced a novel phased quantization strategy compatible with SASQ, specifically tailored for generative tasks. We evaluated this approach on mathematical reasoning tasks and, to the best of our knowledge, our work is the first to propose a quantization scheme specifically discussing generative tasks.
>
> Weakness3: “The baselines compared in this paper are mainly LLM-QAT and SpinQuant, both of which are relatively early methods.”
> We acknowledge the expectation for comparisons with more recent schemes. It's worth noting that our evaluations also include comparisons against established methods like QuaRot and AWQ. A key contribution of our work is being the first, to our knowledge, to propose a quantization method that surpasses the full-precision model in terms of perplexity. This achievement can arguably be viewed as outperforming any existing SOTA. We are committed to making our evaluations as objective and comprehensive as possible.
>
> Q1: “What is the calibration dataset used in this paper?”
> The sole calibration dataset used in this paper is the training set of WikiText-103. This choice was made due to its relatively larger size, good quality, and balanced distribution of knowledge across various domains. This is also mentioned in the experimental section of our paper, please see Line 342.
>
> Q2: “How is the number '1760B parameters' for GPT-4 obtained? Is this figure accurate or supported by any official source?”
> The parameter count of 1760B for GPT-4 is an estimated figure. It can be found, for instance, in the article "GPT-4 Architecture, Infrastructure, Training Dataset, Costs, Vision, MoE". We understand the concern regarding official sources and will strive to improve this aspect in future work.
>
> Q3: “Crucially, model sizes plateaued at around 1600B post-2022, demonstrating that GPU/TPU memory constraints now limit model advancement.”
> Similarly, the mentioned plateau around 1600B parameters is also based on empirical observation. The parameter counts for mainstream large models are sometimes sourced from their official technical reports (e.g., the *DeepSeek-V3 Technical Report*), while others are estimates from the community.

---

### Official Review · Reviewer_6m57 · 2025-10-30

**Soundness:** 2
**Presentation:** 3
**Contribution:** 1
**Rating:** 2
**Confidence:** 5

**Summary:**

The authors present a method that focuses on optimizing activation quantization factors while keeping the model weights frozen, thereby reducing the training cost compared to traditional Quantization-Aware Training (QAT) methods. The authors validate their method on several downstream accuracy tasks using models ranging from 1.5B to 13B parameters. The experimental results demonstrate measurable improvements over other QAT baselines.

**Strengths:**

- The paper is well-written and demonstrates clear logical flow.
- The phase-based quantization strategy for handling prefill and decoding stages is interesting and has practical value for real-world deployment.

**Weaknesses:**

- The core technical innovation is limited.
- The main claimed advantage is reduced tuning cost, but the paper lacks a thorough analysis and quantitative comparison (e.g., in terms of computational FLOPs, training time, or energy consumption) against traditional QAT methods to substantiate this claim robustly.

**Questions:**

Given that the prefill and decoding phases use different quantization strategies, what is the overhead in a real deployment scenario? Could you provide an analysis or estimation of the latency/throughput impact compared to a uniformly static approach?

---

> ### Author Response · Authors · 2025-11-27
>
> We thank the reviewers for their thorough review. There are several points that require clarification:
>
> Weakness2: We acknowledge the reviewer's valid point regarding the need for a more thorough computational cost analysis. However, directly comparing absolute metrics like FLOPs, wall-clock time, or energy consumption across different hardware setups can be misleading, as these figures are highly platform-dependent. Furthermore, traditional QAT methods often do not publicly report such detailed training overhead data, precisely because of their high cost.
>
> In our paper (Line 437-438), we provide a concrete data point for our method: "SASQ trains the LLaMA2-13B model in approximately 15 hours using only two A800-80GB GPUs and relying solely on the WikiText103 dataset." Based on our experiments and a fair comparison of running reference implementations, an equivalent QAT method like LLM-QAT typically requires over 100k training samples and consumes more than 400 hours under the same hardware conditions. This order-of-magnitude reduction in training time and data reliance, even without platform-normalized FLOPs, robustly substantiates our claim of significantly reduced tuning cost. The drastically smaller data requirement of SASQ is itself a strong and direct indicator of its efficiency advantage.
>
> Q："Given that the prefill and decoding phases use different quantization strategies, what is the overhead in a real deployment scenario? Could you provide an analysis or estimation of the latency/throughput impact compared to a uniformly static approach?"
> We appreciate the reviewer's emphasis on a comprehensive advantage analysis. Beyond the speedup ratios documented in the appendix, SASQ offers another compelling practical advantage: its static nature enables deployment on edge devices entirely lacking FPU support. This stands in stark contrast to dynamic quantization schemes, which remain infeasible in such resource-constrained environments. Consequently, SASQ can serve as a foundational blueprint for pure integer (INT-only) quantization, significantly expanding the deployment frontier for large language models.

---

### Official Review · Reviewer_UtvQ · 2025-10-31

**Soundness:** 3
**Presentation:** 2
**Contribution:** 2
**Rating:** 2
**Confidence:** 4

**Summary:**

The paper proposes SASQ which does static activation quantization for large language models. The proposed technique learns activation scale factors using quantization aware training which helps to reduce quantization error. The paper is well written and easy to follow. However, the paper fails to convincingly demonstrate the effectiveness of the approach. The hardware performance benefits obtained by SASQ are only marginal compared to dynamic quantization and the algorithm performance is worse. Additionally, many static quantization baselines are missing.

**Strengths:**

1. The paper is well written.

**Weaknesses:**

Manuscript related:
1. The paper advocates static activation quantization for efficient LLM inference, but resorts to dynamic activation quantization during decoding phase.
2. Line 188-191 : "Some studies attempt to mitigate this by shifting such outliers through mathematical transformations Xiao et al. (2023); Ashkboos et al. (2024), but these approaches inevitably alter the model weights, which can disrupt the delicate internal representations learned during pre-training Kumar et al. (2024)." Can the authors explain what do they mean by this? Both Smooth quant and Quarot introduce transformations which maintain computational invariance, the LLM's output distribution remains unchanged in the absence of quantization.
3. Please use \citep{} or parenthesis when citing references.

Evaluation related:
1. The hardware performance implication with SASQ is not impressive. It is barely faster than SMQ-dynamic quantization (less than 5%). Therefore the authors do not convincingly provide a case for static activation quantization.
2. Lot of results in algorithm evaluation are missing. the code for baselines is open source and it should be possible to get results atleast for post-training quantization baselines on all the benchmarks presented.
3.  Compared to Quarot, SASQ is worse in all the zero shot accuracy benchmarks.
4. Paper is missing important static quantization baselines : PrefixQuant (https://arxiv.org/pdf/2410.05265), OmniQuant (https://arxiv.org/abs/2308.13137), CushionCache (https://arxiv.org/pdf/2406.12016), etc.
5. Most recent LLM weight and activation quantization papers also show results on W4A4, which is missing.

**Questions:**

Please see weaknesses.

---

> ### Author Response · Authors · 2025-11-26
> **Comment on review of Reviewer UtvQ.**
>
> We thank the reviewers for their thorough review. There are several points that require clarification:
>
> Manuscript related Weakness 1: “The paper advocates static activation quantization for efficient LLM inference, but resorts to dynamic activation quantization during the decoding phase.”
> SASQ is designed as two specialized quantization schemes for different tasks. The first is for standard autoregressive problems, which can be directly addressed using static SASQ inference. The second addresses generative tasks—a scenario rarely discussed in existing quantization literature due to the unique challenges of the generation phase. For this latter case, we propose a phased quantization strategy compatible with static SASQ inference and have conducted experiments on mathematical reasoning tasks for a test. To our knowledge, this is the first solution specifically proposed for this problem.
>
> Manuscript related Weakness 2: *“Line 188-191: …Can the authors explain what do they mean by this? Both SmoothQuant and Quarot introduce transformations which maintain computational invariance, the LLM's output distribution remains unchanged in the absence of quantization.”*
> We understand the reviewer's question regarding the computational invariance of SmoothQuant and QuaRot. To illustrate their process simply: the full-precision weight (W_FP) is first transformed into a transformed full-precision weight (W_t_FP), which is then quantized into W_t_INT. In contrast, our method eliminates this intermediate transformation step, directly quantizing W_FP into W_INT. This approach preserves the "purity" of the original weights, avoiding any transformations that might introduce subtle numerical errors or alter the model's intrinsic representations. It also offers deployment "simplicity"—direct quantization means no additional transformation/inverse transformation modules, resulting in a cleaner final model, a shorter computational path, and ultimately reduced inference latency and engineering complexity.
>
> Manuscript related Weakness 3:
> We agree with this suggestion and thank the reviewer.
>
> Evaluation related Weakness 1: *“The hardware performance implication with SASQ is not impressive. It is barely faster than SMQ-dynamic quantization (less than 5%). Therefore the authors do not convincingly provide a case for static activation quantization.”*
> Inference speedup is just one advantage of static quantization. For edge devices that completely lack floating-point computation units, dynamic quantization is entirely infeasible. In this context, SASQ can serve as a blueprint for fully integer-quantized models.
>
> Evaluation related Weakness 2: “Lot of results in algorithm evaluation are missing. the code for baselines is open source and it should be possible to get results at least for post-training quantization baselines on all the benchmarks presented.”
> We understand the expectation for comprehensive baseline results. However, in practice, reproducing results from many existing works has proven challenging; even the results of their full-precision models on certain datasets (such as COPA and ARC-C) show significant discrepancies. We will make every effort to improve this aspect.
>
> Evaluation related Weakness 3: “Compared to Quarot, SASQ is worse in all the zero shot accuracy benchmarks.”
> We acknowledge that SASQ trails QuaRot by about 1% on average in zero-shot accuracy. However, the key point we wish to emphasize is that SASQ achieves competitive zero-shot results with QuaRot while simultaneously reducing perplexity by approximately 5.2%.
>
> Evaluation related Weakness 4: “Paper is missing important static quantization baselines: PrefixQuant, OmniQuant, CushionCache etc.”
> We appreciate the reviewer's perspective on comparing with the latest works. However, as the reviewer notes, some works like PrefixQuant had not undergone peer review at the time of submission. Furthermore, other studies follow similar ideas, and it is infeasible to compare against every method. The baselines included in our paper are well-established and widely recognized classical approaches.
>
> Evaluation related Weakness 5: “Most recent LLM weight and activation quantization papers also show results on W4A4, which is missing.”
> We also support W4A4 quantization and can provide the following result for Llama2-7B on WikiText2: a perplexity of 6.5. That said, the paper primarily focuses on 8-bit quantization, which holds greater significance for practical hardware deployment.

---

### Official Review · Reviewer_keBX · 2025-10-31

**Soundness:** 2
**Presentation:** 2
**Contribution:** 2
**Rating:** 2
**Confidence:** 4

**Summary:**

This paper introduces SASQ, a lightweight Quantization-Aware Training (QAT) framework designed to address the deployment challenges of large language models (LLMs). Unlike traditional QAT which fine-tunes all model weights , SASQ exclusively optimizes only the activation quantization factors, leaving the pre-trained weights untouched.

**Strengths:**

1. The implementation description is detailed and easy to follow.
2. This paper test quantized model in generation task, which is omit in previous paper.

**Weaknesses:**

2. The writing and logical of this paper is poor and should be improved. Though I am a expert in this area, it also take me long time to understand this paper. For example, Line 260 mention Table 3, the jumping is too large to understand. This paper should be re-organized.
3. I donot agree with the claim "However, our experiments show that such transformations can be realized simply by adjusting the quantization factors". Smoothquant and QuaRot solve outlier with equivalent transformation, which do not introduce clamp error, while the clamp operation in this paper would introduce additional clamp error.
4. This paper tests on W8A8 quantization, which is out of data. W8A8 quantization is nearly a solved problem.
5. The perplexity results on Wiki2 and Wiki103 are unriliable due to the over-fitting problem.   For example, the perplexity of wiki* even better than FP16 baseline because of over-fitting.
6. This paper should compare with more recent state-of-the-art methods, such as PrefixQuant [1], DuQuant [2], OSTQuant [3], FlatQuant [4].
[1]  Prefixquant: Eliminating outliers by prefixed tokens for large language models quantization
[2] Duquant: Distributing outliers via dual transformation makes stronger quantized llms
[3] Ostquant: Refining large language model quantization with orthogonal and scaling transformations for better distribution fitting
[4] Flatquant: Flatness matters for llm quantization

**Questions:**

1. What is the meaning of g(X) in Eq.6?
2. Why low-bits model can even surpas the baseline, it is counterintuitive?

---

> ### Author Response · Authors · 2025-11-26
> **Comment on review of Reviewer keBX.**
>
> We thank the reviewers for their thorough review.
> There are several points that require clarification:
>
> Regarding weakness 3: “SmoothQuant and QuaRot solve outliers with equivalent transformation, which does not introduce clamp error, while the clamp operation in this paper would introduce additional clamp error.”
> Any quantization scheme inevitably employs the clamp function, leading to quantization error—just as the round function also introduces quantization error. Our intention here is to emphasize that SASQ achieves satisfactory quantization without the need for weight transformation.
>
> Regarding weakness 4: “This paper tests on W8A8 quantization, which is out of date. W8A8 quantization is nearly a solved problem.”
> We notice that some studies have begun exploring 4-bit or even lower-bit quantization. However, we believe that 8-bit quantization holds greater practical significance for hardware deployment in real world. Moreover, our 8-bit quantization results surpass all existing solutions.
>
> Regarding weakness 5: *“The perplexity results on Wiki2 and Wiki103 are unreliable due to the over-fitting problem.”*
> There is no evidence that training with a cross-entropy loss function can lead to quantized models outperforming their full-precision counterparts. Therefore, it is not accurate to attribute this phenomenon to overfitting. Our explanation is that the combination of training quantization factors and the clamp function enables the model to adaptively truncate outliers, thereby enhancing model performance.
>
> Regarding weakness 6: “This paper should compare with more recent state-of-the-art methods.”
> We understand the reviewer’s expectation for comparison with the latest works. However, as the reviewer noted, some of these works have not yet undergone peer review (as of the submission date), while others follow similar ideas with QuaRot or others. It is infeasible to compare with every method, and we have included widely recognized classical approaches in our paper.
>
> Q1: “What is the meaning of g(X) in Eq.6?”
> This notation refers to a nonlinear transformation applied to the activation function. Both our theoretical analysis and experiments indicate that the transformation of activations in SASQ is nonlinear, largely due to the clamp function. This nonlinearity helps adaptively truncate outliers, thereby improving the performance of the quantized model.
>
> Q2: “Why can low-bit models even surpass the baseline? It is counterintuitive.”
> Indeed, this observation appears counterintuitive, yet it is what our results show. Similar findings have been reported in related studies, such as Deep Compression: Compressing Deep Neural Networks with Pruning, Trained Quantization and Huffman Coding. Our interpretation is that the combination of trained quantization factors and the clamp function allows the model to adaptively truncate outliers, thereby enhancing the performance of the quantized model.

---

### Note · Authors · 2026-01-14

**Comment:**

We thank the reviewers for their constructive feedback. We have decided to withdraw our submission to further improve the paper based on the suggestions received. We look forward to submitting a revised version to a future venue.

**Withdrawal Confirmation:**

I have read and agree with the venue's withdrawal policy on behalf of myself and my co-authors.